# LOGAussian: Efficient Local Gathering for Online Feed-forward 3DGS

## Abstract

Feed-forward 3D Gaussian Splatting (3DGS) has attracted increasing attention for its broad applicability and real-time inference capabilities. Despite recent progress through advanced backbones, prevailing pipelines remain offline: concatenating per-view, pixel-aligned splats introduces redundancy and, under streaming input, accumulates errors over time. To address this, we propose **LOGAussian** (LOcal GAthering Gaussians), a lightweight post-hoc module that maintains an incrementally updated and render-ready global 3D Gaussian representation from sequential posed images without per-scene optimization. The approach integrates global scene context and models local correlations among predicted Gaussians, explicitly accommodating sparse-view inputs with depth noise and geometric imprecision. With about 1% additional parameters, our approach yields a compact, consistent set of splats while maintaining or improving rendering quality as the stream progresses.

## 1 Introduction

Representing 3D scenes as a radiance field has become a widely adopted paradigm for novel view synthesis. One central topic in this task is efficiency, which has driven the evolution from Neural Radiance Fields (NeRF) (Mildenhall et al., 2021) to 3D Gaussian Splatting (3DGS) (Kerbl et al., 2023), and more recently, to feed-forward 3DGS models (Charatan et al., 2024; Chen et al., 2024b).

Feed-forward 3DGS directly predicts a splat-based scene representation from a few posed images in a single pass, avoiding per-scene optimization. These methods achieve real-time inference, but currently remain *offline*: they assume a fixed input set and degrade under streaming data. This limitation is especially problematic in robotics and autonomous navigation, where an agent incrementally observes its environment and needs to build a render-ready 3D model on the fly. Real-time novel view synthesis from newly captured frames can guide further exploration and planning, enabling robots or AR/VR systems to localize or move in unseen areas. Existing feed-forward 3DGS pipelines, however, lack such online update mechanisms and thus cannot fully support these time-critical, interactive tasks.

This motivates the key question of our work: *Can a feed-forward 3DGS pipeline be made on-line—incrementally incorporating new views?* Achieving this requires overcoming two challenges. First, per-frame predictions should be fused into a consistent global model. Some concurrent advances (Li et al., 2025) leverage short-term correspondences (e.g., over adjacent frames) to mitigate fragmentation; however, without true global fusion they cannot verify or correct drift and inconsistencies. Second, the splats predicted by feed-forward networks are often noisy or imprecise, especially in the absence of reliable 3D priors (which are typically available only in controlled settings, e.g., indoor scenes (Wang et al., 2024b)). These inaccurate predictions require dedicated pruning and refinement mechanisms in more general scenes.

We propose to maintain a compact, consistent set of Gaussians by explicitly modeling local correlations among Gaussians from incoming frames and the existing global model. However, learning relationships directly in 3D is challenging due to the irregular, unstructured nature of the splat set. To address this, we draw inspiration from feed-forward pipelines that exploit view-aligned representations—such as image-based cost volumes (Chen et al., 2024b) or canonical coordinate spaces (Wang et al., 2024a; Ye et al., 2024)—and introduce a projection-based attention module. By projecting all Gaussians onto a 2D image plane and processing them in a structured order, our approach can effectively detects overlapping or inconsistent splats. This design enables merging

redundant primitives and pruning outliers on the fly, substantially improving the consistency of the accumulated 3D model.

Building on these insights, we propose **LOGAussian** (LOcal GAthering Gaussians), an online feed-forward 3DGS pipeline that incrementally fuses per-frame splats into a global model. After each frame, LOGAussian produces a refined set of Gaussians and merges them into the global model. Our fusion module is lightweight and adds negligible overhead to the backbone, while explicitly modeling inter-splat relationships to maintain coherence. Consequently, **LOGAussian** substantially reduces redundant splats and limits error accumulation in an online setting, bringing feed-forward 3DGS closer to practical real-time scene capture.

To sum up, our contributions are three-fold:

- We propose a lightweight post-hoc module for *online feed-forward 3DGS* that enables incremental novel view synthesis on streaming inputs.

- We introduce a *projection-based attention module* to capture local correlations among Gaussians, even with imprecise or noisy predictions.

- Leveraging local correlation-aware features, we introduce (i) an importance score-based *pruning strategy* to eliminate redundant Gaussians, and (ii) a *refinement module* that updates Gaussians, producing a compact set of splats while preserving rendering quality on the fly.

- We validate the effectiveness of our method on a state-of-the-art backbone, DepthSplat (Xu et al., 2025), demonstrating that our design can serve as a *plug-in module* for feed-forward 3DGS pipelines—substantially reducing redundancy and error accumulation without per-scene optimization.

## 2 RELATED WORK

**Sparse-view novel view synthesis.**    Neural rendering has rapidly advanced the field of novel view synthesis (NVS) with both implicit (Mildenhall et al., 2021) and explicit (Kerbl et al., 2023) scene representations. Classical methods typically require dense multi-view images and extensive training time for every new scene. To avoid costly per-scene optimization, recent research has focused on generalizable, feed-forward approaches that directly predict 3D representations from a few images by learning priors from large-scale datasets. Existing feed-forward methods include generalizable NeRF variants (Yu et al., 2021; Chen et al., 2021) and 3DGS-based approaches (Charatan et al., 2024; Chen et al., 2024b; Min et al., 2024; Wewer et al., 2024), achieving photorealistic NVS with known camera poses of input views. Their extensions explore generative refinement (Chen et al., 2024c), coarse-to-fine hierarchy (Tang et al., 2024a; Zhang et al., 2025), or strong depth priors (Xu et al., 2025) for improved synthesis.

Following the feed-forward 3D reconstruction pipeline of DUSt3R (Wang et al., 2024a), recent works (Smart et al., 2024; Ye et al., 2024) propose to directly append Gaussian heads to the backbone for novel view synthesis, achieving fast and generalizable NVS from unposed images. On the other hand, InstantSplat (Fan et al., 2024) uses feed-forward 3DGS as initialization for fast per-scene optimization, while MV-DUSt3R (Tang et al., 2024b) and FASt3R (Yang et al., 2025) extend the feed-forward reconstruction pipeline to multi-view scenarios, avoiding pair-wise alignment.

Despite their efficiency and improved rendering quality, feed-forward 3DGS models often suffer from redundancy and degraded quality due to overlapped splats induced by naive concatenation of per-view pixel-aligned splats, which gets worse when the input views become denser. Our work addresses this issue by introducing a lightweight pipeline that learns local and fine-grained relationships among nearby splats, effectively pruning redundant splats and refining the remaining ones for better compactness and quality.

**Online 3D Gpaaussian Splatting.**    Extending 3DGS from static image collections to sequential input introduces new challenges of redundancy, drift, and scalability. Recent attempts process streaming frames by finding correspondences between consecutive frames (Li et al., 2025; Leroy et al., 2024). This yields fast updates but lack long-term context. Other geometry-guided designs (Wang et al., 2024b) focus on indoor videos with reliable geometry. These methods rely on depth alignment and pixel-level fusion to prune overlaps, showing effectiveness on scenes with reliable depth cues.

However, methods that depend on geometric alignment suffer from imprecise or noisy estimations in more general, unconstrained scenes. Our approach takes a different path: we fuse splats directly in view space, where even noisy depth predictions preserve local correlations. By modeling inter-splat relationships through attention and pruning, we build an online feed-forward 3DGS pipeline that maintains a compact, any-time renderable representation. This design avoids heavy geometric assumptions while maintaining both efficiency and fidelity over long sequences.

**Locality-Aware Gaussian Splatting.** 3D Gaussians naturally exhibit strong spatial coherence—neighboring splats often share similar geometry, color, or semantics. Per-scene optimization works have leveraged this locality for better reconstruction and compression (Lu et al., 2024; Chen et al., 2024a). In the feed-forward regime, locality is mostly addressed post-hoc. SplatFormer applies a point transformer over initial splats to correct attributes and enhance robustness to out-of-distribution views Chen et al. (2025). Generative Densification inserts a lightweight generator that hallucinates fine-scale splats conditioned on coarse splats, densifying the representation in one forward pass Nam et al. (2024). However, these approaches still treat splats from different views independently, leaving substantial duplication unaddressed.

## 3 PRELIMINARIES

**3D Gaussian Splatting (Kerbl et al., 2023).** In 3DGS, a static scene is represented by a collection of anisotropic 3D Gaussians, $\{G^{(i)}\}_{i=1}^N$. Each 3D Gaussian is parameterized by its spatial mean $\boldsymbol{\mu} \in \mathbb{R}^3$, covariance matrix $\boldsymbol{\Sigma} \in \mathbb{R}^{3\times3}$, opacity $\alpha$, and color $\boldsymbol{c}$. During rendering, 3D Gaussians are projected onto the image plane through an efficient and differentiable rasterization pipeline. The photometric loss $\mathcal{L}(\mathcal{I}, \mathcal{I}^{gt})$ between the rendered image $\mathcal{I}$ and the ground truth $\mathcal{I}^{gt}$ is applied to optimize 3D Gaussians' parameters.

**Feed-forward 3DGS (Chen et al., 2024b; Xu et al., 2025).** Feed-forward 3DGS replaces iterative optimization with a single-pass network. Recent promising methods usually leverage strong geometric priors (Xu et al., 2023; 2022; Yang et al., 2024) learned from large datasets to produce a good initialization. This paradigm treats most intermediate outputs as pixel-aligned dense 2D maps. Specifically, given $V$ calibrated input images $\{\boldsymbol{I}^{(i)}\}_{i=1}^V$ ($\boldsymbol{I}^{(i)} \in \mathbb{R}^{H\times W\times 3}$) with corresponding camera poses $\{\boldsymbol{P}^{(i)}\}_{i=1}^V$ ($\boldsymbol{P}^{(i)} \in \mathbb{R}^{3\times4}$), the feed-forward 3DGS pipeline first extracts per-view multiview-aware feature maps $\boldsymbol{F}^{(i)} \in \mathbb{R}^{H\times W\times C_f}$, where $C_f$ is the feature dimension. Then, for each view $i$, the pipeline uses the corresponding feature map to predict a dense pixel-aligned depth map $\boldsymbol{D}^{(i)} \in \mathbb{R}^{H\times W}$, which is unprojected into a 3D point map $\mathbf{X}^{(i)} \in \mathbb{R}^{H\times W\times 3}$. A lightweight *Gaussian head* then takes each feature map as input and outputs per-pixel Gaussian attributes to form the "Gaussian map", $\mathcal{G}^{(i)} \in \mathbb{R}^{H\times W\times C_g}$, where $C_g$ is the total dimension of Gaussian parameters. Finally, all per-view "Gaussian maps" are concatenated to a 3D Gaussian set of the dimension $V \times H \times W \times C_g$, by taking their union, $\mathcal{G} = \bigcup_{v=1}^V \mathcal{G}^{(v)}$, which supports novel view synthesis.

## 4 METHOD

We propose a lightweight post-hoc LOcal GAthering network, termed **LOGAussian**, that brings online processing capabilities to an offline feed-forward 3DGS backbone. Feed-forward 3DGS instantiates Gaussians by per-pixel regression without modeling inter-Gaussian relations or considering global context. This often produces duplicate splats in overlapping regions and propagates per-view noise—leading to redundancy, error accumulation, and a lack of global consistency. LOGAussian addresses these issues by locally grouping Gaussians (Sec. 4.1), applying patch-wise self-attention to capture local correlations (Sec. 4.2), and performing a lightweight global fusion (Sec. 4.3) to maintain an any-time renderable scene representation.

### 4.1 LOCAL GATHERING NETWORK

Given a stream of posed image frames $\{\mathbf{I}^{(t)}, \mathbf{P}^{(t)}\}$, our method maintains and incrementally updates a renderable Gaussian scene $\mathcal{G}$ for each frame in a single feed-forward pass. As illustrated in Fig. 1, LOGAussian operates in two stages for each incoming frame: a *local fusion* stage that

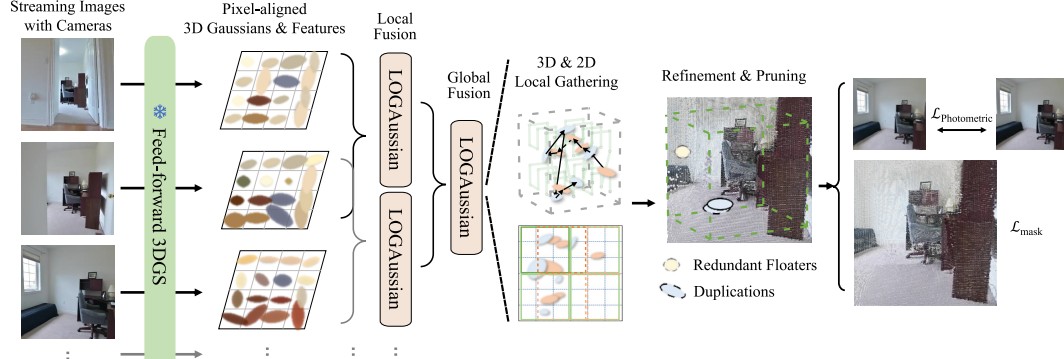

Figure 1: **Overview of our method.** Our LOcal GAthering (LOGAussian) network serves as a post-hoc module to enable online feed-forward 3DGS. It takes both pixel-aligned 3D Gaussians and their features as input and operates in two stages: one local fusion stage that refines the immediate multi-view input, and a global fusion stage that merges the accumulated scene representation. LOGAussian first partitions the Gaussians into spatially localized patches using efficient 3D serialization and 2D projection, and applies self-attention within each patch to capture local context. A refinement and pruning head then removes redundant Gaussians and updates the retained. The network is supervised with a photometric loss for image quality and a mask loss to encourage sparsity.

refines the immediate multi-view input, and a *global fusion* stage that merges the result with the accumulated scene representation. Both stages share the same network architecture and weights to ensure consistency, efficiency, and ease of training.

**Initialization.** For clarity, we first consider the simplest online scenario of two frames ($t = 2$) to illustrate our approach and will later generalize to longer sequences in Sec. 4.3. When a new frame $I^{(t)}$ arrives, along with the previous frame $I^{(t-1)}$, the feed-forward 3DGS pipeline processes these two views to produce an initial dense set of pixel-aligned Gaussians $\mathcal{G}^{(t)} = \{ G_k = (\boldsymbol{\mu}_k, \boldsymbol{\Sigma}_k, \alpha_k, \mathbf{c}_k) \}_{k=1}^{2 \times H \times W}$ and their associated features $\mathcal{F}^{(t)} = \{\mathbf{f}_k\}_{k=1}^{2 \times H \times W}$, where each feature vector $\mathbf{f}_k \in \mathbb{R}^{C_f}$. Our objective is to compress this dense prediction into a compact set of Gaussians and features of the size ($N < 2HW$) in a single forward pass, while preserving rendering quality.

Modern feed-forward 3DGS pipelines (e.g., DepthSplat (Xu et al., 2025)) leverage learned multi-view or monocular depth cues to produce dense per-pixel depth estimates. The resulting Gaussians and their associated deep features (from large-scale pretrained encoders (Yang et al., 2024; Xu et al., 2023)) provide a strong initialization for refinement. Whereas the backbone focuses on regressing pixel-aligned predictions from features, the proposed approach instead learns to exploit local correlations within this dense field of Gaussians.

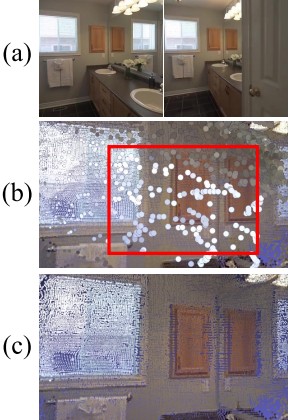

Figure 2: **Challenge of 3D serialization.** (a) Input context images. (b) Dispersed predicted 3D Gaussian. (c) Reduced set of 3D Gaussians from our method.

**3D serialization.** To efficiently group Gaussians by spatial proximity, we employ a 3D serialization strategy (Wu et al., 2024), such as the Morton (Z-order) curve. Specifically, a locality-preserving ordering of the Gaussian centers is computed using a 3D space-filling curve that maps each 3D center $\boldsymbol{\mu}$ to a 1D index $z = \psi(\boldsymbol{\mu})$. In this mapping, the local spatial relationships are roughly preserved as illustrated in Fig. 3(c). We then sort all Gaussians by $z$ and partition the sorted sequence into non-overlapping segments of length $P$, yielding $\lceil N/P \rceil$ local patches. The entire process of encoding, sorting, and partitioning maintains real-time performance thanks to the well-studied serialization algorithm (Wu et al., 2024).

**Projected 2D partitioning.** While 3D serialization preserves approximate 3D spatial locality, it can be suboptimal in sparse-view settings. Specifically, due to limited view coverage and occlusions, the reconstructed 3D points often exhibit *uneven density* and *inconsistent geometry* across views. For instance, depth estimates can vary significantly between viewpoints due to prediction errors and scale variations, leading to dispersed 3D Gaussian distributions as Fig. 2(b) illustrates. Although these estimations diverge in 3D, their projections onto the image plane may remain adjacent, offering opportunities for aggregation and comparison in 2D.

To compensate for the shortage of 3D space-filling, we propose a projected 2D partitioning strategy. Concretely, for each camera view, all Gaussian centers are projected onto that view's image plane (Fig. 3(b)). Gaussians outside the camera frustum are discarded, and the remaining projected points are grouped based on their 2D image coordinates. This view-aware grouping strategy ensures that patches correspond to observable regions in the image and inherently respects occlusions, producing more coherent local patches aligned with the visual content.

Compared to traditional depth-based fusion methods (Wang et al., 2024b; Sun et al., 2021), the proposed approach avoids explicit depth comparisons, which are often unreliable in the presence of noise or inconsistent estimates. Instead, integrating information in the image-space domain other than on 3D proximity helps prevent the accumulation of depth errors. This strategy is akin to using multiple projections as in locality-sensitive hashing (Andoni et al., 2015) to ensure spatial neighbors are likely grouped in at least one projection. The learned features and local attention mechanism implicitly reason about visibility and redundancy, producing a consistent global scene model.

## 4.2 HYBRID ARCHITECTURE OF THE LOCAL GATHERING MODULE

The LOGAussian module $\theta_{\text{LOGA}}$ uses a hybrid architecture that interleaves 2D and 3D grouping layers within a feed-forward network. As illustrated in Fig. 3(d), each predicted Gaussian and its features is first embedded into a latent vector via a shared two-layer MLP. All Gaussians are then partitioned into local patches using alternating 2D projections and 3D serializations, and a stack of local self-attention layers is applied to capture fine-grained correlations among Gaussians.

**Hybrid partitioning.** In practice, each 2D grouping stage performs $V$ sequential 2D-based grouping operations (one per input view). Gaussians are projected into a particular view and partitioned into patches as described above, after which multi-head self-attention (Vaswani et al., 2017) is applied within each patch. A shifted-window attention mechanism (Liu et al., 2021) is employed to mitigate edge effects from this fixed patch partitioning. Gaussians not observed in the current view (i.e., those outside the frustum) are carried forward via a skip connection to be processed in later stages. After processing all $V$ views, a 3D serialization layer is applied to capture any remaining local interactions in the full 3D space. During training, random variations of the serialization order are used (Wu et al., 2024) to avoid introducing bias.

**Local self-attention.** Within each spatially grouped patch, a multi-head self-attention (Vaswani et al., 2017) layer is applied to aggregate contextual information among the Gaussian features. This operation refines the feature descriptors by allowing each Gaussian to attend to its neighbors, thereby capturing fine-grained correlations useful for subsequent redundancy reduction and attribute refinement. Formally, for a patch of $P$ grouped Gaussians with feature vectors $\{\tilde{F}^{(p)}\}_{p=1}^P$, the attention is computed as:

$$\text{MultiHead}(\boldsymbol{Q}, \boldsymbol{K}, \boldsymbol{V}) = \text{Concat}(\text{head}_1, \ldots, \text{head}_h)\,\boldsymbol{W}_o, \tag{1}$$

$$\text{head}_i = \text{Softmax}\Big(\frac{\boldsymbol{Q}_i \boldsymbol{K}_i^T}{\sqrt{d_i}}\Big)\,\boldsymbol{V}_i, \tag{2}$$

$$\boldsymbol{Q}_i = \boldsymbol{F}\,\boldsymbol{W}_q^{(i)}, \quad \boldsymbol{K}_i = \boldsymbol{F}\,\boldsymbol{W}_k^{(i)}, \quad \boldsymbol{V}_i = \boldsymbol{F}\,\boldsymbol{W}_v^{(i)}, \tag{3}$$

Here $\boldsymbol{F}$ stands for the aggregated feature matrix, $\boldsymbol{W}_*$ are the learned projection matrices used in attention, and $d_i$ is the feature dimensionality of head $i$.

**Refinement and Score-based Masking Heads** To eliminate duplicates and refine the remaining Gaussians, two lightweight heads are attached to the LOGAussian network: (i) a score-based *mask*

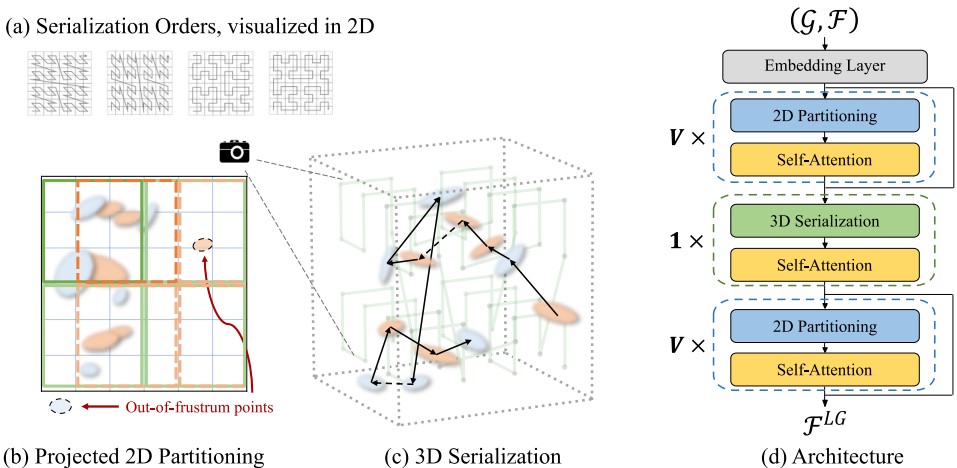

Figure 3: **Overview of our method components.** (a) Illustration of four types of serialization orders in 2D (source: Wu et al. (2024)), where each encodes different spatial relationships. During training, we randomly choose one order to partition our Gaussians to avoid serialization bias. (b) A projected 2D partitioning strategy with the shifted-window attention. Compared with the 3D serialization in (c), 2D projection on camera planes aligns well with the sparse-view settings and is robust to imprecise 3D geometries and drifted depths. (c) The overall architecture of our local gathering module, composed of stacked 2D and 3D partitioning and local self-attention, where $V$ stands for the number of input views.

*head* for predicting which Gaussians to retain, and (ii) a *refinement head* for updating the parameters of the kept Gaussians. Both heads operate on the locality-aware features produced by the network.

Formally, given the enhanced features from the LOGAussian backbone $\theta_{\text{LOGA}}$, we predict per-Gaussian mask scores and residual updates as:

$$m = \text{MaskHead}\big(\theta_{\text{LOGA}}(\{\mathcal{G}, \mathcal{F}\})\big), \tag{4}$$

$$\Delta\mathcal{G} = \text{GaussianHead}\big(\theta_{\text{LOGA}}(\{\mathcal{G}, \mathcal{F}\})\big), \tag{5}$$

$$\tilde{\mathcal{G}} = \{\, G_i + \Delta G_i \mid m_i > \tau \,\}, \tag{6}$$

where $m_i \in [0, 1]$ denotes the retention probability, $\Delta G_i$ is the predicted residual update for each Gaussian, and $\tau$ is a threshold controlling retention. Gaussians with $m_i > \tau$ are updated and kept in the set $\tilde{\mathcal{G}}$, while those with lower scores are pruned.

Both heads are implemented as small two-layer MLPs. The $\text{MaskHead}$ applies a sigmoid activation to produce $m_i$, and a straight-through estimator (STE) (Bengio et al., 2013) is adopted to enable gradient backpropagation through the hard thresholding operation. The $\text{GaussianHead}$ predicts residual offsets for all Gaussian attributes. The refined feature descriptors produced by the LOGAussian backbone are also preserved for use in the subsequent global fusion stage.

### 4.3 GLOBAL FUSION FOR MULTI-VIEW INPUTS

Over time, the feed-forward pipeline can accumulate errors in the scene representation, since it lacks explicit global consistency constraints and is usually trained without direct metric depth supervision. Past global fusion approaches that rely on raw 3D alignment or depth comparisons thus often neglect this drift. To address this, we introduce a learned global fusion step analogous to our local fusion.

When a new frame $I^{(t)}$ arrives, we first apply the local fusion network to $I^{(t-1)}$ and $I^{(t)}$, obtaining a compact set of Gaussians $\mathcal{G}_{\text{local}}^{(t)}$ for the new frame. This set is then concatenated with the existing global scene $\mathcal{G}_{\text{global}}^{(t-1)}$ to form $\mathcal{G}_{\text{combined}}^{(t)} = \mathcal{G}_{\text{global}}^{(t-1)} \cup \mathcal{G}_{\text{local}}^{(t)}$. Finally, $\mathcal{G}_{\text{combined}}^{(t)}$ is fed through the LOGAussian network, yielding an updated global model $\mathcal{G}_{\text{global}}^{(t)}$. Rather than updating the global model on every single frame, which would be computationally costly, the global fusion is performed periodically (every $M$ frames) using a sliding window of the most recent $M$ frames. This strategy

allows the model to prune redundant points and refine the scene using both local and broader context, limiting error accumulation.

**Training details.** We adapt an existing offline feed-forward 3DGS method, DepthSplat (Xu et al., 2025) as our backbone and extend it to the online setting. During training, we initially focus on the two-view scenario, which is the minimum requirement for online reconstruction, and then progressively increase the number of input views once the two-view model has converged. To stabilize optimization, we employ a strategy of alternating the mask head on and off, inspired by iterative schemes in per-scene optimized 3DGS methods. Specifically, we disable the mask pruning branch for a fixed number of iterations, allowing the network to refine all Gaussians without pruning, and then re-enable it for the next set of iterations to learn pruning. This cycle is repeated every $K$ iterations. Such an alternating schedule prevents the model from overly discarding Gaussians and empirically leads to better convergence for both heads.

Our loss function consists of a photometric loss and a sparsity regularizer:

$$\mathcal{L}_{\text{loss}} = \mathcal{L}_{\text{photo}} + \lambda_{\text{mask}}\mathcal{L}_{\text{mask}}, \tag{7}$$

where $\mathcal{L}_{\text{photo}} = 1.0 * \mathcal{L}_{\text{MSE}} + 0.05 * \mathcal{L}_{\text{LPIPS}}$ measures the photometric error on novel views, and $\mathcal{L}_{\text{mask}} = \frac{1}{N}\sum m_i$ with $\lambda_{\text{mask}} = 10^{-4}$ encourages sparsity on masks. Here $N$ is the number of input Gaussians and $m_i$ is the mask score for Gaussian $i$.

## 5 EXPERIMENTS

### 5.1 EXPERIMENTAL SETTINGS

**Baselines.** Since no publicly available online feed-forward 3D Gaussian Splatting (3DGS) methods exist, we compared our approach with several offline feed-forward 3DGS baselines to demonstrate its effectiveness. We included (i) the backbone model, DepthSplat Xu et al. (2025) to measure the impact of our enhancements; (ii) MVSplat (Chen et al., 2024b), a multi-view Gaussian splatting model that uses a plane-sweep cost volume for geometry learning; and (iii) Gaussian Graph Network (GGN) (Zhang et al., 2024), an enhanced feed-forward 3DGS method focusing on efficiency. We also implemented two naive online variants of DepthSplat for comparison: (iv) DepthSplat-O, which incrementally adds new Gaussians from incoming frames, and (v) DepthSplat-OR, which additionally applies the depth rescaling technique to mitigate depth drifting in online scenarios. These two online baselines highlighted the contributions of our proposed LOGAussian module. All methods were evaluated under identical data protocols and metrics to ensure fair comparison.

**Datasets & Evaluation.** We conducted experiments on two public benchmarks for novel view synthesis: RealEstate10K (RE10K) Zhou et al. (2018) and DL3DV Ling et al. (2024), both comprising real-world video segments with known camera poses. We followed DepthSplat Xu et al. (2025) to split training, test, and validation sets and pre-process data. On RE10K, we evaluated using input image sequences of 4, 8, and 50 views. The 4- and 8-view configurations match those of the GGN for fair comparisons, while the 50-view "long sequence" setting was included to demonstrate our method's robustness on challenging, extended sequences. Cross-dataset generalization was evaluated on DL3DV using models trained on RE10K. Following community practice Mildenhall et al. (2021), we reported novel view synthesis metrics of PSNR, SSIM, and LPIPS. We measured the total number of Gaussians (denoted as "#" in the tables) to quantify redundancy reduction. We further evaluated end-to-end efficiency by measuring inference time, rendering FPS, and peak GPU memory.

**Implementation Details.** Our method was implemented in PyTorch and trained on four NVIDIA A100 GPUs for one day. We adopted the small variant of the DepthSplat model as the backbone and kept its parameters frozen during training. Baseline results were either taken from reported values or reproduced using publicly released codes. Further details are included in the appendix.

### 5.2 RENDERING PERFORMANCE AND GAUSSIAN REDUCTION

**Short sequence results.** Table 1 presents a comparison between our method and baselines on RE10K using 4 and 8 image sequences as inputs. Our method outperforms all baselines by a

Table 1: **Comparisons with feed-forward 3DGS methods on RE10K. Top**: offline methods with access to all inputs; **Bottom**: online methods that process frames incrementally. Our approach achieves the best PSNR/SSIM and lowest LPIPS, with reduced Gaussians compared to the backbone.

| Method | 4 Views | | | | 8 Views | | | |
|---|---|---|---|---|---|---|---|---|
| | PSNR ↑ | SSIM ↑ | LPIPS ↓ | # ↓ | PSNR ↑ | SSIM ↑ | LPIPS ↓ | # ↓ |
| MVSplat | 23.91 | 0.885 | 0.141 | 262 K | 21.39 | 0.846 | 0.183 | 524 K |
| DepthSplat | 26.56 | 0.910 | 0.113 | 262 K | 24.03 | 0.875 | 0.145 | 524 K |
| GGN | 24.76 | 0.784 | 0.172 | 102 K | 25.15 | 0.793 | 0.168 | 126 K |
| DepthSplat-O | 26.73 | 0.916 | 0.107 | 262 K | 24.27 | 0.881 | 0.140 | 524 K |
| DepthSplat-OR | 25.37 | 0.877 | 0.130 | 262 K | 22.47 | 0.810 | 0.186 | 524 K |
| Ours | 28.76 | 0.928 | 0.096 | 183 K | 28.15 | 0.918 | 0.106 | 285 K |

Table 2: **Long-sequence evaluation on RE10K (50 views).** Offline baselines (MVSplat, GGN) run out of memory (OOM) on 50-view inputs.

| Method | PSNR ↑ | SSIM ↑ | LPIPS ↓ | # ↓ |
|---|---|---|---|---|
| DepthSplat | 19.48 | 0.706 | 0.255 | 3277K |
| DepthSplat-O | 19.96 | 0.727 | 0.246 | 3266K |
| DepthSplat-OR | 18.34 | 0.641 | 0.302 | 3265 K |
| Ours | 23.43 | 0.810 | 0.181 | 1070K |

Table 3: **Cross-dataset generalization** (RE10K to DL3DV) with 12 image sequences of high resolution (448×256).

| Method | PSNR ↑ | SSIM ↑ | LPIPS ↓ | # ↓ |
|---|---|---|---|---|
| MVSplat | 15.34 | 0.413 | 0.548 | 1375 K |
| DepthSplat | 17.61 | 0.571 | 0.404 | 1371 K |
| Ours | 18.40 | 0.608 | 0.371 | 964 K |

significant margin in PSNR, SSIM, and LPIPS, despite operating in an online manner. Moreover, our method effectively reduces the number of Gaussians compared to the backbone, demonstrating its capability to prune redundant splats while preserving rendering quality. Notably, simply applying depth rescaling (DepthSplat-OR) does not reliably improve the performance over the naive online version (DepthSplat-O), showing PSNR changes of –1.36, -1.8, and –1.62 dB across 4-, 8-, and 50-view settings. This highlights the limitations of purely geometric fixes in handling drift and redundancy issues in an online setting. Corresponding qualitative comparisons are included in Fig. 4.

**Long sequence results.** We further evaluate on long, dense input sequences (50 views), which pose heavy memory and accumulated errors challenges. Offline methods like MVSplat and GGN struggle with memory overflow (OOM). While DepthSplat can process 50 views by building cost volumes only from adjacent frames, this lack of global context causes a severe drop in quality: it produces many unrecoverable floaters (artifacts of spurious Gaussians), as presented in Fig. 4 and Tab. 2. Moreover, this design also requires knowledge of all input frames beforehand, not suiting the online scenario. In contrast, our method handles long sequences by explicitly fusing each incoming frame's prediction into a global model. This incremental global fusion corrects accumulated errors and eliminates floaters.

In practice, our approach can be combined with key-frame selection strategies, such as ZPressor Wang et al. (2025). We construct an offline variant of LOGAussian by applying pose-based farthest-point camera sampling while keeping our local gathering and fusion modules. As Tab. 4 illustrates, with access to all camera poses, our offline variant achieves comparable rendering quality to ZPressor while using only about 70% of its Gaussian count. This shows that LOGAussian is compatible with ZPressor-style key-frame selection in the offline regime, while our main contribution targets the online setting where such pose-privileged strategies are not available.

Table 4: **Extending to the offline variant (50 views).** By adopting farthest-point sampling, our offline variant achieves rendering quality comparable to ZPressor while using fewer Gaussians.

| Method | PSNR ↑ | SSIM ↑ | LPIPS ↓ | # ↓ |
|---|---|---|---|---|
| Ours (Online) | 23.43 | 0.810 | 0.181 | 1070k |
| ZPressor | 26.75 | 0.888 | 0.111 | 393k |
| Ours (Offline) | 26.73 | 0.893 | 0.119 | 278k |

Table 5: **Integration with MVSplat Chen et al. (2024b).**

| Method | PSNR | SSIM | LPIPS | # |
|---|---|---|---|---|
| MVSplat (4 views) | 23.91 | 0.885 | 0.141 | 262K |
| + Ours | 27.82 | 0.896 | 0.125 | 176K |
| MVSplat (8 views) | 21.39 | 0.846 | 0.183 | 524K |
| + Ours | 25.70 | 0.854 | 0.164 | 310K |
| MVSplat (50 views) | OOM | – | – | – |
| + Ours | 20.96 | 0.707 | 0.261 | 1371K |

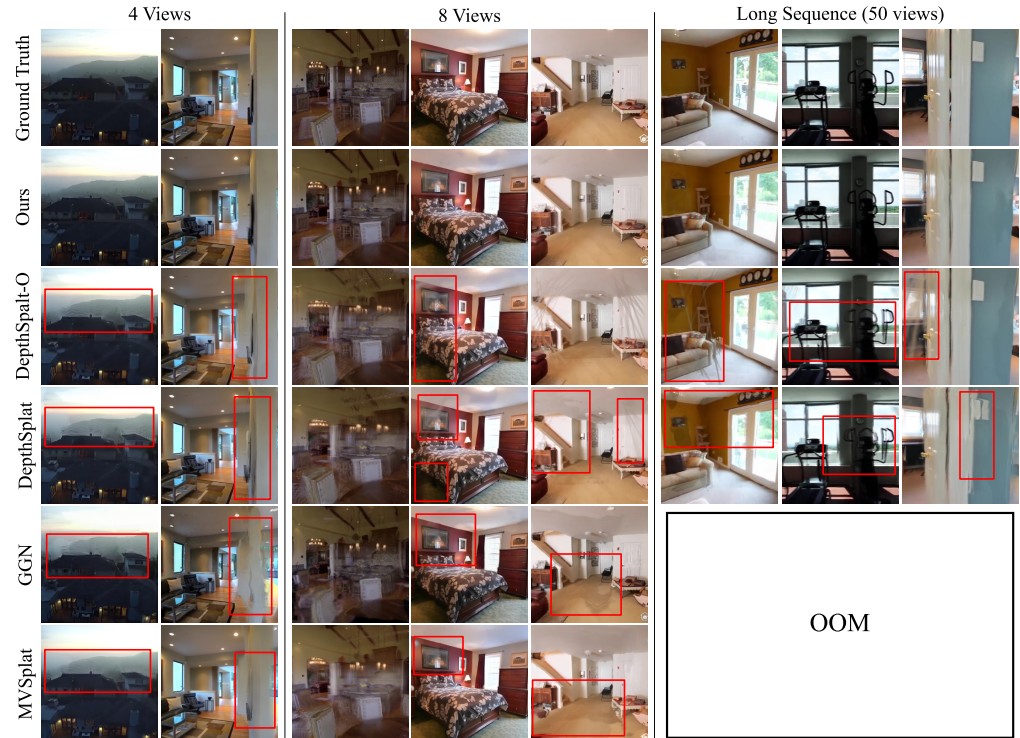

Figure 4: **Qualitative comparisons on streaming inputs.** Insets mark regions where offline baselines and naive accumulation methods fail. Our online feed-forward 3DGS mitigates the floating and duplicate splats issue, producing sharper edges and fewer ghosting artifacts over time. For long image sequences, offline methods including MVSplat and GGN run out of the memory (OOM).

Table 6: Inference time, peak GPU memory, and FPS under different numbers of input views.

| Method | 4 Views | | | 8 Views | | | 50 Views | | |
| --- | --- | --- | --- | --- | --- | --- | --- | --- | --- |
| | Time (ms) | Mem (GB) | FPS | Time (ms) | Mem (GB) | FPS | Time (ms) | Mem (GB) | FPS |
| MVSplat | 102.70 | 2.48 | 460 | 218.13 | 4.90 | 368 | OOM | – | – |
| DepthSplat | 71.11 | 4.77 | 489 | 117.97 | 9.25 | 394 | 576.56 | 56.29 | 167 |
| DepthSplat-O | 39.14 | 1.70 | 462 | 45.36 | 2.06 | 398 | 51.35 | 9.11 | 171 |
| Ours | 75.35 | 2.82 | 531 | 86.74 | 3.32 | 455 | 63.25 | 7.25 | 298 |

**Generalization to another feed-forward 3DGS model.** LOGAussian is backbone-agnostic and can be seamlessly integrated into existing feed-forward 3DGS pipelines. When combined with MVSplat Chen et al. (2024b) (Tab. 5), it consistently improves quality and reduces Gaussians for 4- and 8-view settings, and remains efficient under 50 views where MVSplat runs OOM. This confirms LOGAussian as an effective, plug-in module for enhancing efficiency and robustness.

**Efficiency evaluation.** We evaluate end-to-end inference time, peak GPU memory, and rendering FPS under 4-, 8-, and 50-view inputs (Tab. 6). Our method consistently achieves the highest FPS and uses substantially less memory than offline DepthSplat, with more favorable scaling as views increase. Compared with DepthSplat-O, it adds only modest overhead in few-view settings and becomes both faster and more memory-efficient at 50 views due to effective pruning. Our method further achieves significantly better rendering quality than all baselines, as reported in Tab. 1 and Tab. 2.

**Cross-dataset generalization.** We test our model's generalization by training on RE10K and evaluating on the DL3DV dataset (12-view input sequences at 448×256 resolution). Importantly, neither the DepthSplat backbone nor our LOGaussian module is exposed to DL3DV data during

training. Following DepthSplat, we used 12 image sequences as inputs and measured NVS rendering quality on 100 frames. As shown in Tab. 3, our method still achieves better performance than baseline methods in the cross-dataset scenario. This demonstrates the generalizability of our approach.

Table 7: **Ablation on sliding-window size and 2D projection-partitioning patch size.** Our default configuration uses a sliding-window size of 4 and a patch size of 4.

| | 8 Views | | | | 50 Views | | | |
|---|---|---|---|---|---|---|---|---|
| Method | PSNR ↑ | SSIM ↑ | LPIPS ↓ | # ↓ | PSNR ↑ | SSIM ↑ | LPIPS ↓ | # ↓ |
| DepthSplat | 24.03 | 0.875 | 0.145 | 524k | 19.48 | 0.706 | 0.255 | 3277k |
| DepthSplat-O | 24.27 | 0.881 | 0.140 | 524k | 19.96 | 0.727 | 0.246 | 3266k |
| *Sliding-window size* | | | | | | | | |
| Ours (Local) | 28.25 | 0.918 | 0.106 | 374k | 23.15 | 0.799 | 0.186 | 1604k |
| Ours ($M = 4$) | 28.15 | 0.918 | 0.106 | 285k | 23.43 | 0.810 | 0.181 | 1070k |
| Ours ($M = 8$) | 28.03 | 0.918 | 0.105 | 314k | 23.20 | 0.804 | 0.183 | 1493k |
| *Patch size* | | | | | | | | |
| Ours (patch size 2) | 28.29 | 0.919 | 0.107 | 292k | 23.49 | 0.809 | 0.184 | 1099k |
| Ours (patch size 4) | 28.15 | 0.918 | 0.106 | 285k | 23.43 | 0.810 | 0.181 | 1070k |
| Ours (patch size 6) | 27.99 | 0.916 | 0.108 | 282k | 23.43 | 0.810 | 0.182 | 1050k |

## 5.3 ABLATION STUDIES.

For the ablation studies, we vary the sliding-window size used in the global fusion and the patch size used in the 2D projected partitioning. We also provide additional analyses on the masking threshold, pruned floaters, and residual updates in the supplementary material.

Our default configuration uses a sliding-window size of $M = 4$. We conduct ablations under the 8- and 50-view settings by comparing: (i) no global fusion ("local"), (ii) window size 4, and (iii) window size 8.We also include the offline and online variants of DepthSplat as references. The results are summarized in Tab. 7. Without global fusion ("local"), the number of Gaussians remains relatively large, which degrades performance on long image sequences. Adding global fusion reduces the Gaussian count and maintains or improves rendering quality, particularly at 50 views. When increasing the window size from 4 to 8, we observe a slight drop in rendering quality and less effective pruning. This behavior aligns with our design assumption that Gaussians exhibit locality: the 2D partition–based self-attention depends on the number of views, and a larger window introduces more complex and less localized fusion.

We ablate the effect of different patch sizes in Tab. 7, where the default configuration uses a patch size of 4. With smaller patch sizes, each patch contains fewer candidate points, resulting in fewer pruning opportunities and therefore a larger number of retained Gaussians. We also observe that smaller patch sizes yield slightly better rendering quality. This suggests that smaller patches group candidates that are closer in view space, allowing the self-attention module to operate on more visually coherent neighborhoods, which can benefit the fused representation.

## 6 CONCLUSION

We propose a post-hoc module, **LOGAussian**, to enable online feed-forward 3DGS. The method addresses two central issues of offline feed-forward 3DGS—redundancy and missing global context—by modeling local inter-splat correlations and fusing information over time. We integrate 2D projection and 3D serialization, localized self-attention, and joint refinement–pruning to produce a compact, any-time renderable scene representation. A shared module is reused for both local (two-view) and periodic global fusion, reducing noisy and redundant splats and improving rendering quality on the fly. Our method is lightweight and operates in a feed-forward way.

**Limitations and Future Work.** As a plug-in module for feed-forward 3DGS pipelines, the performance of our method is inherently dependent on the quality of the underlying 3DGS backbone. Our LOGAussian does not specify how poses are obtained and assumes known poses. Additionally, extending our 2D projection scheme to settings with inaccurate or noisy camera poses remains an open challenge and will be our future work.

**Reproducibility statement.**    Model architecture and training protocols are summarized in Section 4.2 and Sec. 5.1. Detailed implementation details are provided in the appendix A.1.

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

## A  APPENDIX

### A.1  IMPLEMENTATION DETAILS.

**3D serialization.**   Our serialized encoding scheme leverages the locality-preserving space-filling curves to define traversal orders over 3D points, following the approach in Wu et al. (2024). Specifically, each point $p \in \mathbb{R}^n$ is first quantized using a uniform grid of size $g \in \mathbb{R}$, producing a discrete index $\lfloor p/g \rfloor \in \mathbb{Z}^n$. A bijective space-filling curve function $\mathcal{F} : \mathbb{Z}^n \to \mathbb{Z}$ is then applied to map these grid indices to unique integer serialization codes, preserving their spatial proximity.

For 3D serialization, the grid size $g$ is chosen based on the resolution used in constructing the cost volume Chen et al. (2024b); Xu et al. (2025). Following Wu et al. (2024), we encode each serialization index using 48-bit integers and employ 16-bit integers for batch processing, sufficient for our settings. This serialization strategy enables efficient local self-attention in 3D space. For a detailed derivation and explanation, please refer to the original paper Wu et al. (2024).

**Implementation Details.**   Our method was implemented in PyTorch and trained on four NVIDIA A100 GPUs for roughly one day. We adopted the small variant of the DepthSplat model as the backbone and kept its parameters frozen during training. We employed the Adam optimizer Kingma (2014) with an initial learning rate of $1 \times 10^{-4}$, following the OneCycleLR annealing schedule (Smith & Topin, 2018). The alternating enabling and disenabling mask iteration $K$ was set to $2,000$ and the mask threshold is set to $0.05$. For serialization and local self-attention, we adopted the implementation from Point Transformer v3 (Wu et al., 2024) and Swin Transformer (Liu et al., 2021). Baseline results were either taken from reported values if available or reproduced using publicly released codes.

## A.2 More ablation studies.

We ablate the mask threshold $\tau$ on 8-view settings. As discussed earlier, our default configuration uses threshold 0.05, and we keep this value fixed throughout training. As Tab.8 illustrates, without the mask ($\tau = 0.0$), the residual updating module alone already improves rendering quality over DepthSplat-O, but it cannot achieve the same balance between quality and compactness as the default setting, since it does not prune any redundant Gaussians, which may also result in overblurring effects. As $\tau$ increases, more Gaussians are pruned: moderate thresholds (e.g., 0.04–0.05) yield strong rendering quality with a substantially reduced number of Gaussians, while overly aggressive thresholds (e.g., 0.055) oversparsify the representation and hurt image quality.

Table 8: Ablation on the mask threshold $\tau$ under 8 input views.

| Method | $\tau$ | PSNR ↑ | SSIM ↑ | LPIPS ↓ | # Gaussians ↓ |
|---|---|---|---|---|---|
| DepthSplat-O | – | 24.27 | 0.881 | 0.140 | 524k |
| Ours | 0.00 | 27.72 | 0.910 | 0.112 | 524k |
| Ours | 0.04 | 28.14 | 0.918 | 0.104 | 326k |
| Ours (default) | 0.05 | 28.15 | 0.918 | 0.106 | 285k |
| Ours | 0.055 | 25.76 | 0.891 | 0.144 | 222k |

We ablate the effect of the residual updates defined in Eq. 5. The residual branch predicts updates for all Gaussian attributes—means, scales, rotations, opacities, and SH colors. Since these attributes reside in different units and are tightly coupled during optimization, comparing their raw residual magnitudes is not very meaningful. Instead, we measure their importance by selectively disabling residual updates for individual attributes and reporting the resulting performance under the 8-view setting (Tab.9).

Table 9: Ablation on the residual updates.

| Method | W/O Updating | PSNR | SSIM | LPIPS |
|---|---|---|---|---|
| DepthSplat-O | – | 24.27 | 0.881 | 0.140 |
| Ours (prune only) | All | 26.23 | 0.904 | 0.125 |
| Ours | Scales | 27.89 | 0.917 | 0.108 |
| Ours | Rots | 28.13 | 0.918 | 0.106 |
| Ours | Opacities | 28.23 | 0.910 | 0.116 |
| Ours | SH | 27.76 | 0.916 | 0.111 |
| Ours | Means | 28.15 | 0.918 | 0.107 |
| Ours (default) | – | 28.15 | 0.918 | 0.106 |

"Ours (prune only)" disables all residual updates and performs only pruning and fusion on top of DepthSplat-O. Rows marked with a specific attribute in the "W/O updating" column disable residual updates only for that attribute while leaving the others intact. "Ours (default)" updates all attributes jointly.

We observe that all configurations with residual updates significantly outperform DepthSplat-O and the "prune only" variant. Relative to the default, disabling scales or SH noticeably degrades PSNR/SSIM and LPIPS, and disabling opacities yields slightly higher PSNR but clearly worse SSIM/LPIPS, indicating that opacity corrections are particularly important for perceptual quality. Disabling means or rotations has only a minor effect (differences < 0.03 dB PSNR and very small SSIM/LPIPS changes), suggesting that these attributes are somewhat more robust to being kept fixed in the residual branch.

We hypothesize that this behavior stems from how the model reallocates coverage after pruning: once redundant Gaussians are removed, the remaining Gaussians need to adjust their spatial extent (scales) and contributions (opacities and colors) to better cover the scene, which makes residual updates on these attributes especially impactful. During training, we always optimize all attributes jointly and do not train attribute-specific models; this ablation is only used post hoc to analyze their relative influence.

We further analyze floater pruning on 10 scenes using 4 input views. For each Gaussian, we compute the gradient norm of the photometric loss with respect to its color parameters, which serves as an indicator of how sensitive the rendered error is to changes in that Gaussian's appearance. We rank Gaussians by this gradient norm and divide them into five bands (top 20%, 20–40%, ..., last 20%). For each band, we report the pruning ratio—the fraction of Gaussians in that band removed by LOGAussian, as summarized in Tab. 10.

We observe that both the top and bottom 20% bands exhibit noticeably higher pruning ratios than the middle bands. Intuitively, the top band corresponds to Gaussians with large gradients, many of which behave as floaters or harmful artifacts due to their strong influence on the photometric error. Conversely, the lowest-gradient band contains Gaussians that contribute little to the reconstruction and are largely redundant. This suggests that LOGAussian tends to remove both highly erroneous and redundant Gaussians, while being more conservative in the middle bands.

Table 10: Pruning ratio across gradient-based importance bands (4 views, averaged over 10 scenes).

| Gradient Band (by color-gradient norm) | Pruning Ratio (%) |
|---|---|
| Top 20% | 28.13 |
| 20–40% | 25.32 |
| 40–60% | 23.80 |
| 60–80% | 25.26 |
| Last 20% | 31.89 |

**Limitations on long image sequences.** We evaluate LOGAussian on sequences of approximately 300 frames and do not observe GPU memory becoming a bottleneck. Unlike offline feed-forward methods, our approach is not directly tied to the sequence length because the locality-preserving pruning and update scheme keeps the active Gaussian set compact throughout processing. For substantially longer or more challenging videos, the main limitation is likely to come from the feed-forward backbones trained under sparse-view regimes. Long sequences often contain unreliable frames—such as textureless regions, large exposure changes, imperfect epipolar geometry, or motion blur—which degrade depth and feature predictions and impact most feed-forward 3DGS pipelines. Combining the backbone-agnostic LOGAussian with more robust 3D feed-forward backbones is an interesting direction for future work to better handle difficult long-sequence scenarios.

Table 11: **Performance under a sequence of roughly 300 frames.**

| Method | PSNR | SSIM | LPIPS | Peak GPU Mem. (GB) | # Gaussians |
|---|---|---|---|---|---|
| DepthSplat-O | OOM | – | – | – | – |
| Ours | 18.08 | 0.725 | 0.306 | 17 | 3959.4K |

## B  MORE QUALITATIVE RESULTS.

**Redundancy reduction.** To highlight the redundancy in the predictions of feed-forward 3DGS and demonstrate the advantages of our method, we design a special visualization strategy: Let $C_i$ denote the input camera $i$, and $\mathcal{G}_i$ represent the set of pixel-aligned Gaussians predicted from $C_i$. We define $\mathcal{R}(\mathcal{G}_i, C_i)$ as the rendering of Gaussians $\mathcal{G}_i$ from the same viewpoint of $C_i$. This *source-view rendering* provides a direct view of the redundancy of predicted Gaussians, enabling a clear comparison before and after pruning. We present the local fusion results on RE10K in Fig. 5, where our method effectively removes redundant Gaussians, while maintaining high rendering quality.

## C  ASSETS LICENSE AND BROADER IMPACTS.

**Broader Impacts** Our methods require training on high-performance GPU(s). The computational resources could possibly cause the global climate change.

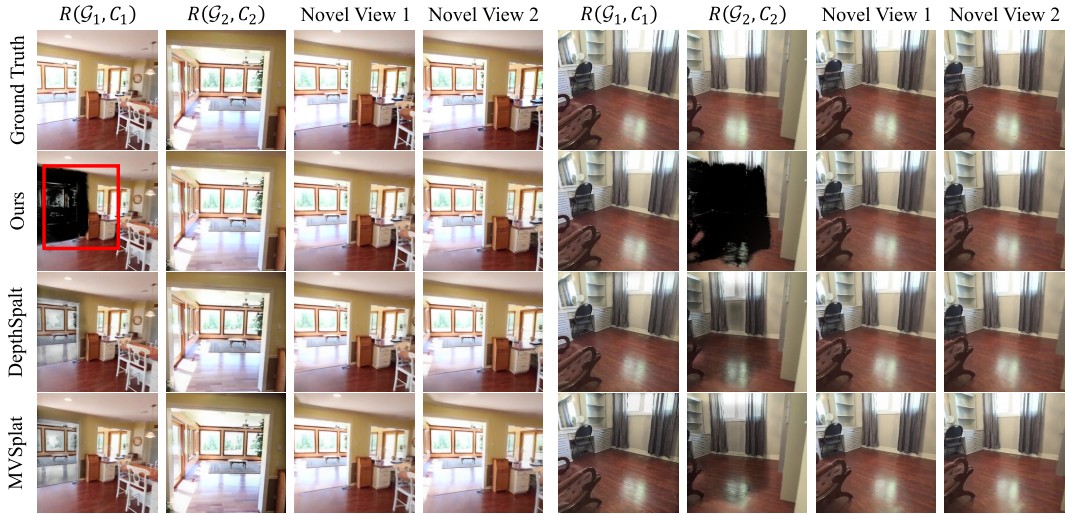

Figure 5: Visualization of redundancy reduction and rendering quality for two-view inference on RealEstate10K. To illustrate the redundancy removal effect of our method, we render each set of Gaussians onto its corresponding source image, denoted by $\mathcal{R}(\mathcal{G}_i, C_i)$. The ground truth row of $\mathcal{R}(\mathcal{G}_i, C_i)$ thus corresponds to the original input images used by the feed-forward 3DGS. Compared with other feed-forward 3DGS methods, our method could remove redundant Gaussians without sacrificing rendering quality. Black regions correspond to "pruned Gaussians".

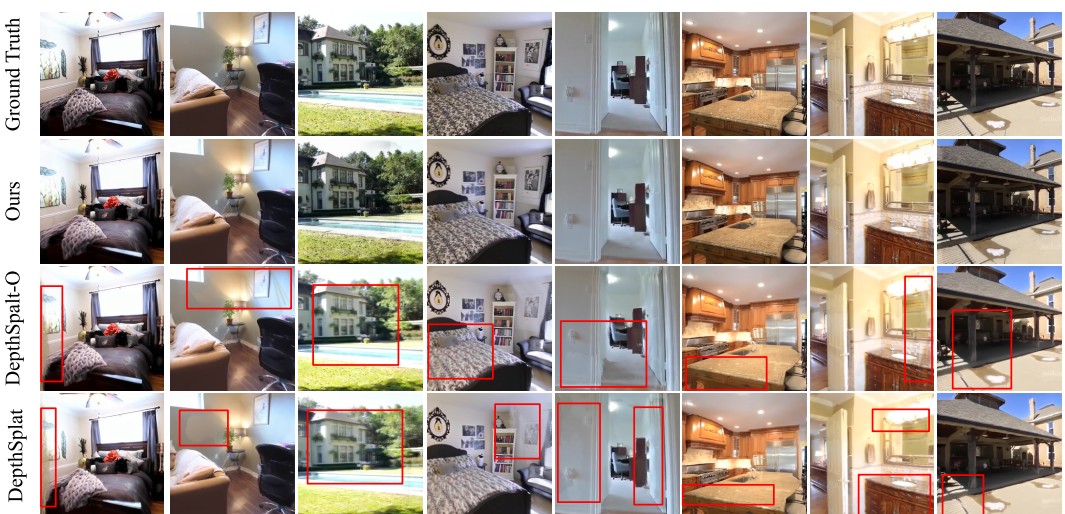

Figure 6: More qualitative results on long image sequences (50 views).

We list the licenses of all the existing assets we have used, including the code and data.

Table 12: License of used assets in this work.

| Asset | License Link |
| --- | --- |
| 3D Gaussian Splatting Kerbl et al. (2023) | https://github.com/graphdeco-inria/gaussian-splatting/blob/main/LICENSE.md |
| Depth Splat Xu et al. (2025) | https://github.com/cvg/depthsplat/blob/main/LICENSE |
| MVSplat Chen et al. (2024b) | https://github.com/donydchen/mvsplat/blob/main/LICENSE |
| Point Transformer V3 Wu et al. (2024) | https://github.com/Pointcept/PointTransformerV3/blob/main/LICENSE |
| PyTorch | https://github.com/pytorch/pytorch/blob/main/LICENSE |
| RealEstate10K Zhou et al. (2018) | https://google.github.io/realestate10k/download.html (CC BY 4.0) |
| DL3DV Ling et al. (2024) | https://github.com/DL3DV-10K/Dataset/blob/main/License.md |

