# OpenReview forum: "LOGAussian: Efficient Local Gathering for Online Feed-forward 3DGS"
_ICLR.cc/2026/Conference — Submitted to ICLR 2026_

### Official Review · Reviewer_axVZ · 2025-10-23

**Soundness:** 2
**Presentation:** 3
**Contribution:** 2
**Rating:** 2
**Confidence:** 4

**Summary:**

This paper proposes LOGAussian, a post-hoc lightweight module designed to enable online processing for feed-forward 3D Gaussian Splatting (3DGS). The authors claim that the module performs local and global fusion of Gaussian splats over streaming frames, using a hybrid 2D projection + 3D serialization strategy with attention-based pruning and refinement. The method is evaluated on RealEstate10K and DL3DV datasets using DepthSplat as the backbone, reporting improved PSNR/SSIM and reduced Gaussian redundancy.

**Strengths:**

1. The paper provides extensive references and a reasonably clear experimental protocol.

2. The idea of integrating 2D projections and 3D serialization is intuitive and might inspire minor future extensions in online fusion.

**Weaknesses:**

1. The main pipeline is a hybrid of DepthSplat’s cost volume and standard patch-based transformer refinement. The proposed local fusion is essentially a masked attention plus pruning head, not conceptually new.

2. Overclaimed efficiency. Despite “lightweight” claims, the model still relies on multi-head attention and repeated fusion steps. No inference-time analysis or FLOP comparison is provided.

3. The method is described as “online,” yet the global fusion is done every M frames, making it quasi-offline.

**Questions:**

1. Can the authors provide exact runtime measurements (FPS) compared to DepthSplat and MVSplat for fair evaluation of “efficiency”?

2. How is the “1% additional parameters” computed—does it include the attention weights and dual MLP heads?

3. Did the authors attempt to train with unposed or partially posed sequences to justify the claimed “online” robustness?

---

> ### Author Response · Authors · 2025-11-25
> **Thank you for your reviews (Part 1 / 2)**
>
> **Q1: The main pipeline is not conceptually new.**
>
> **A1:** We agree that the individual building blocks we use — the DepthSplat backbone, transformer-based refinement, and masked attention — are not new by themselves, and we do not claim novelty at the operator level. Our contribution lies in how these components are integrated into a **feed-forward 3DGS fusion module** specifically tailored to the **online, sparse-view** setting.
>
> Technically, we aim to design a module that suits sparse views and noisy 3D predictions from feed-forward backbones, while preserving locality. To this end, we introduce a **view-space–aligned projection and patchifying scheme**: we project Gaussians into the image plane, group them into view-aligned patches, and perform within-patch self-attention over projected Gaussians before importance-based fusion. This design enables fine-grained control over Gaussians (e.g., capturing and handling floaters) and is naturally suited to noisy feed-forward predictions. To our knowledge, such a pipeline has not been explored in prior feed-forward 3DGS methods.
>
> By contrast, existing approaches rely on **global cross-attention** or large receptive fields that do not explicitly encode locality and are expensive in memory and compute. One concurrent work, ReSplat, shares a similar spirit of fine-grained control and uses recurrent KNN-based attention over Gaussians; however, it requires explicit KNN search to establish spatial neighborhoods and is not specifically designed around the sparse-view setting. Our design makes a different choice: we encode locality directly in view space via within-patch self-attention over projected Gaussians, which both respects the geometric structure of 3DGS and leads to the favorable runtime/memory behavior reported in Table A.
>
> In this sense, while the underlying components are standard, their specific organization for **online feed-forward 3DGS fusion under sparse views** is, to our knowledge, new and practically meaningful. Meanwhile, online processing for feed-forward 3DGS remains relatively underexplored, and our method takes a step toward this direction.
>
> **Q2: Overclaimed efficiency; Exact runtime measurements compared to DepthSplat and MVSplat.**
>
> **A2:** We provide explicit runtime and memory measurements in the general response Table A. Compared to **DepthSplat** (offline), our method consistently uses much less GPU memory and achieves higher FPS, especially as the number of views grows (e.g., 50-view setting: 56.29 GB vs **7.25 GB**, 167 FPS vs **298 FPS**). Compared to **DepthSplat-O**, our method introduces modest extra cost at 4/8 views due to additional pruning and fusion, but already achieves higher rendering FPS at 4 views and becomes both **faster** and **more memory-efficient** at 50 views. Across all settings, we are more efficient than **MVSplat**, which also runs OOM at 50 views while our method remains stable. We believe these measurements directly address and mitigate the concern of "overclaimed" efficiency.

---

> ### Author Response · Authors · 2025-11-25
> **Thank you for your reviews (Part 2/2)**
>
> **Q3: Online vs Quasi-offline.**
>
> **A3:** We would like to clarify that the distinction between **online** and **offline** here comes from whether inference uses only information available at the current time step, rather than from the use of a sliding window. For example, ZPressor [A] is clearly **offline**: its farthest-sampling–based keyframe selection requires access to **all** input views in advance. In contrast, an **online** algorithm processes inputs sequentially and makes decisions (including rendering) without future knowledge.
>
> Our LOGAussian setting is strictly online: the fusion operates on a sliding window over **past** frames only, with no access to future frames. The window size controls temporal context, not offline access to the full sequence. Sliding-window designs are standard in online 3D/vision systems and are well aligned with the locality property of 3D reconstruction tasks. An example is visual SLAM: Modern SLAM pipelines perform local bundle adjustment on a sliding window of recent frames to maintain short-term consistency. They may also trigger global bundle adjustment (or loop-closure optimization) over all previously seen frames to correct drift. Despite these periodic global refinements, SLAM is universally regarded as a strictly online system, because it never waits for future frames, and the current state (pose + map) is always immediately available. Our setting matches this definition: Local fusion is performed every frame, while global fusion is an amortized refinement over past frames only, analogous to global BA in SLAM. No step requires access to future observations, and rendering can be done at each time-step without delay.
>
> For completeness, we also show that LOGAussian can be used in an **offline** manner, with further enhanced performance. In Table D, we apply a ZPressor-style farthest-sampling strategy to select keyframes, while keeping our local and global fusion modules unchanged.
>
> **Table D. Using LOGAussian in an offline manner (50 views).**
> | Method | PSNR | SSIM | LPIPS | # Gaussians |
> |--|--|--|--|--|
> | Ours (Online)   | 23.434 | 0.810 | 0.181 | 1070k |
> | ZPressor | 26.754 | 0.888 | 0.111 | 393k  |
> | Ours (Offline)  | 26.733 | 0.893 | 0.119 | 278k |
>
> The offline variant of LOGAussian achieves **comparable rendering quality** to ZPressor while using **fewer Gaussians**, showing that LOGAussian naturally extends to the offline regime, but our primary contribution is in the **online / incremental** setting where future information is not available.
>
> [A] Wang, Weijie, et al. "ZPressor: Bottleneck-Aware Compression for Scalable Feed-Forward 3DGS." NeurIPS 2025.
>
> **Q4: How is the "1% additional parameters" computed—does it include the attention weights and dual MLP heads?**
>
> **A4:** The "1% additional parameters" is computed using `torch.numel` over all trainable parameters of the LOGAussian module, **including** the attention weights and both MLP heads. The backbone model consists of roughly "38M" parameters, while our added module consists of roughly "385K" parameters.
>
> **Q5: Did the authors attempt to train with unposed or partially posed sequences to justify the claimed "online" robustness?**
> **A5:** We agree that our current method is a post-hoc module built on top of a posed feed-forward backbone and therefore inherits its assumptions, including the availability of camera poses. Designing a fully feed-forward pipeline that estimates poses on-the-fly from purely RGB inputs, while simultaneously maintaining high-quality 3DGS reconstruction, is an active and challenging research problem, and is closely related to recent progress in 3D foundation models (e.g., VGGT [A], DepthAnything V3 [B]). We consider this direction complementary to our work rather than the main focus of this paper.
>
> We would like to highlight that our LOGAussian itself is backbone-agnostic, and the module itself does not specify how poses are obtained and does not modify the pose-estimation process. Once an unposed or partially posed backbone provides per-frame camera parameters and 3DGS, LOGAussian can, in principle, be applied on top in the same way. We will further clarify this limitation in the paper and view combining LOGAussian with unposed 3D backbones as an interesting direction for future exploration.
>
> [A] Wang, Jianyuan, et al. "Vggt: Visual geometry grounded transformer." Proceedings of the Computer Vision and Pattern Recognition Conference. 2025.
> [B] Lin, Haotong, et al. "Depth Anything 3: Recovering the Visual Space from Any Views." arXiv preprint arXiv:2511.10647 (2025).

---

### Official Review · Reviewer_1oHa · 2025-10-30

**Soundness:** 3
**Presentation:** 3
**Contribution:** 3
**Rating:** 6
**Confidence:** 4

**Summary:**

This paper aims at fusing redundant 3DGS in existing feed-forward 3DGS methods using a learnable network. Unlike previous feed-forward methods, which produces abundant 3DGS in a pixel-aligned way. To do this, it a lightweight module which incrementally updates and refine global 3DGS from sequential images. The network updates 3DGS of an incoming frame alongside the global 3DGS via a local self-attention mechanism. The network is trained solely on the RE10K dataset and evaluated on the RE10K and DL3DV10K test set. Experiments show that the baseline (DepthSplat) performance is largely improved in both the sparse view and dense view settings.

**Strengths:**

(1) The related work and method parts are well-written and easy to understand.

(2) The incremental updating strategy is effective to remove redundant 3DGS, and the performance is greatly improved -- compared to DepthSplat.

(3) The proposed method can be used as a plug-in to many other feed-forward 3DGS methods, which is an important contribution to the community.

**Weaknesses:**

- There are some formatting issues:
(1) A "." is omitted at Line309: "offsets for all Gaussian attributes The refined feature ..."

(2) The qualitative comparisons of "Visualization of the challenging floaters issue" is included into Table 2 which makes the formatting very strange. It is recommended to arrange it into an individual figure.

(3) At line 468, Table 4 is not referenced correctly.

- Though the progressive updating strategy is effective in removing 3DGS, its performance in handling long sequences is unclear: when the number of 3DGS grows, it will needs longer time and more GPU memories to fuse/update 3DGS, which could limit its application in longer videos.

- The experiment part lacks some important baselines which are also related to this work, e.g., ZPressor [1], Long-LRM [2]

[1] zpressor: bottleneck-aware compression for scalable feed-forward 3dgs. NeurIPS 2025
[2] Long-LRM: Long-sequence Large Reconstruction Model for Wide-coverage Gaussian Splats. ICCV 2025

**Questions:**

I am curious about the performance of the method in full sequence videos/images. I would appreciate and raise my score if the authors can provide additional experiments on this.

---

> ### Author Response · Authors · 2025-11-25
> **Thank you for your reviews**
>
> **Q1: Formatting issues.**
>
> **A1:** We thank the reviewer for pointing these out. We will fix all these issues in the final version.
>
> **Q2: Missing baselines (ZPressor, Long-LRM).**
>
> **A2:** Below we provide a direct comparison with ZPressor on the RealEstate10K dataset with 50 views. For Long-LRM, we note that the released models are trained on datasets and with backbones that are not directly aligned with ours, making a controlled comparison difficult. We also would like to emphasize that both ZPressor and Long-LRM are proposed for an **offline** setting.
>
> For ZPressor, the farthest-view sampling and clustering strategy requires access to all camera poses of the sequence **in advance**, which is not compatible with the online scenario we target. We also would like to highlight that our method is agnostic to backbone and could be easily integrated with this design. To provide a fair reference, we construct an offline variant of LOGAussian: we adopt pose-based farthest-point sampling over cameras while keeping our local gathering and fusion module. Both ZPressor and our method use DepthSplat as the backbone for fair comparison. The results are shown in Table D:
>
> **Table D. Comparison with ZPressor on RE10K (50 views)**
> | 50 Views    |  PSNR    |  SSIM  |  LPIPS   |  # Gaussians   |
> |--|--|--|--|--|
> | Ours - Online  |  23.434 | 0.810 |  0.181  | 1070k |
> | ZPressor  |  **26.754** | *0.888* |  **0.111**  | *393k* |
> | Ours - Offline  |  *26.733* | **0.893** |  *0.119*  | **278k** |
>
> With access to all camera poses (offline setting), our offline variant achieves **comparable rendering quality** to ZPressor while using **fewer Gaussians** (about 70% of ZPressor's count). This shows that LOGAussian can be combined with ZPressor-style keyframe selection in the offline regime, while our main contribution focuses on the online / incremental setting where such pose-privileged strategies are not available. We will add this table (and corresponding discussion) to the paper to support our claim on lines 419–420 that our method can be combined with keyframe selection methods.
>
> **Q3: (a) Performance in handling long sequences; (b) With longer time and more GPU memories to fuse/update 3DGS.**
>
> **A3:** Our running memory is governed by the number of active Gaussians in the global representation, not by the sequence length. With the locality-preserving pruning and updating scheme, and by running self-attention on projected Gaussians within view-space patches (rather than global cross-view attention), the size of this global set stays compact. In our current experiments, we have tested sequences of roughly **300 frames** (Table G) and did not observe GPU memory becoming the bottleneck, unlike the offline methods.
>
> The main difficulty on very long sequences comes from the feed-forward backbone (trained under sparse views), rather than from the fusion module itself: long sequences often contain unreliable frames (e.g., textureless regions, exposure shifts, motion blur), under which depth and feature predictions degrade and all feed-forward 3DGS pipelines are affected. Since LOGAussian is backbone-agnostic, it can in principle be combined with more robust backbones, and we will clarify in the paper that our present experiments demonstrate stable memory/runtime behavior up to a few hundred frames, while reconstruction quality on very long sequences is limited, sourcing from the backbones.
>
>
> **Table G. Performance under roughly 300 frames.**
> | | PSNR | SSIM | LPIPS | Peak GPU memory (GB) | # Gaussians |
> |--|--|--|--|--|--|
> | DepthSplat | OOM | | | | |
> | Ours | 18.0847 | 0.725 | 0.306 | 17 | 3959.4K | |
>
> We also provide runtime and memory measurements in the general response Table A. Compared to **DepthSplat** (offline), our method consistently uses much less GPU memory and achieves higher FPS, especially as the number of views grows (e.g., 50-view setting: 56.29 GB vs **7.25 GB**, 167 FPS vs **298 FPS**). Compared to **DepthSplat-O**, our method introduces modest extra cost at 4/8 views due to additional pruning and fusion, but already achieves higher rendering FPS at 4 views and becomes both **faster** and **more memory-efficient** at 50 views. Across all settings, we are more efficient than **MVSplat**, which also runs OOM at 50 views while our method remains stable.

---

### Official Review · Reviewer_MysD · 2025-10-31

**Soundness:** 3
**Presentation:** 1
**Contribution:** 2
**Rating:** 4
**Confidence:** 5

**Summary:**

This paper introduces LOGAussian (LOcal GAthering Gaussians), a lightweight post-hoc module that enables online and incremental processing for feed-forward 3D Gaussian Splatting (3DGS). LOGAussian leverages 3D serialization and 2D projection to model local correlations among Gaussians, followed by an importance score–based pruning and refinement module to update or remove redundant global splats. The proposed module adds only about 1% additional parameters to DepthSplat, while significantly improving both rendering quality and computational efficiency on the RealEstate10K and DL3DV datasets.

**Strengths:**

(1) Well-motivated problem: The paper clearly identifies a key limitation of existing feed-forward 3DGS methods, which operate offline and cannot handle sequential or streaming inputs.

(2) Simple and effective design: The proposed approach introduces a mask head for filtering and a residual prediction head for updating Gaussian attributes, offering a straightforward yet efficient way to maintain compact and consistent scene representations.

(3) Strong experimental results: The method achieves consistent quantitative improvements over DepthSplat across multiple datasets and varying input-view configurations.

**Weaknesses:**

The reviewer finds the authors’ approach interesting and potentially impactful. However, the presentation lacks important implementation details, making it difficult for readers to fully grasp the proposed method.

(1) The paper introduces a threshold τ to generate the Gaussian mask, which determines how many Gaussians are retained. This hyperparameter is crucial for understanding pruning behavior, yet the authors do not provide its default value or any ablation study analyzing its effect.

(2) The authors mention that the global model is not updated on every frame but instead uses a sliding window of the most recent M frames. Since M directly affects inference speed and rendering quality, an ablation study and the default setting should be reported.

(3) The paper does not report inference time, which is essential for evaluating the claimed efficiency. Additionally, reporting peak GPU memory usage would help quantify the computational overhead more precisely.

(4) The proposed method appears incompatible with unposed-image settings such as NoPoseNoProblem [1], where the first frame serves as the canonical space. LOGAussian depends on the base model to process each new incoming view, which may limit its applicability in unposed or pose-free scenarios.

[1]  No Pose, No Problem: Surprisingly Simple 3D Gaussian Splats from Sparse Unposed Images (ICLR2025)

**Questions:**

(1) Can LOGAussian handle sequences with thousands of views?

(2) Can LOGAussian be integrated with MVSplat, and if so, how does it perform in that setting?

---

> ### Author Response · Authors · 2025-11-25
> **Thank you for your reviews. (Part 1/2)**
>
> **Q1: Ablation studies on the mask threshold**
>
> **A1:** We have added an ablation on the mask threshold $\tau$ in Table E on 8-view settings. Our default configuration uses **$\tau$ = 0.05**, and we keep this value fixed throughout training.
>
> Without the mask ($\tau$ = 0.0), the residual updating module alone already improves rendering quality over DepthSplat-O, but it cannot achieve the same balance between quality and compactness as the default setting, since it does not prune any redundant Gaussians, which may also result in overblurring effects. As $\tau$ increases, more Gaussians are pruned: moderate thresholds (e.g., 0.04–0.05) yield strong rendering quality with a substantially reduced number of Gaussians, while overly aggressive thresholds (e.g., 0.055) oversparsify the representation and hurt image quality.
>
> **Table E. Ablation on the mask threshold $\tau$ (8 views).**
>
> | Method| $\tau$ | PSNR  | SSIM   | LPIPS  | # Gaussians |
> |-|-|-|-|-|-|
> | DepthSplat-O    | – | 24.27 | 0.881 | 0.140 | 524k |
> | Ours | 0.00  | 27.72 | 0.910 | 0.112 | 524k |
> | Ours | 0.04  | *28.14* | **0.918** | **0.104** | 326k |
> | Ours (default)  | 0.05  | **28.15** | **0.918** | *0.106* | *285k* |
> | Ours | 0.055 | 25.76 | 0.8911 | 0.144 | **222k** |
>
> We will add this table and explicitly state the default $\tau$ in the paper to clarify the pruning behavior and robustness to this hyperparameter.
>
> **Q2 & Q3: Ablation on the sliding window size; inference time and memory usage.**
>
> **A2 & A3:** Thank you for these suggestions. We have added the ablations on the sliding window size and the detailed efficiency analysis (inference time, FPS, and peak GPU memory) in the general response A1 and below Table B, and will incorporate them into the revised paper.
>
> Our default configuration uses a sliding window size of **M = 4**. We conduct ablations under 8- and 50-view settings, comparing: (i) no global fusion ("Only Local Fusion"), (ii) window size 4, and (iii) window size 8. We also include the offline and online variants of DepthSplat as references. Results are summarized in Table B.
>
> Without global fusion ("Only Local Fusion"), the number of Gaussians remains relatively large, which degrades performance on long image sequences. Adding global fusion reduces the Gaussian count and maintains or improves rendering quality, especially at 50 views. When increasing the window size from 4 to 8, we observe a slight rendering quality drop and less effective pruning. This aligns with our design assumption that Gaussians exhibit locality: the 2D partitioning–based self-attention depends on the number of views, and a larger window makes the fusion more complex and less localized.
>
> **Table B. Ablation on sliding window size M.**
> | 8 Views | PSNR   | SSIM  | LPIPS | # Gaussians |
> |--|--|--|--|--|
> | DepthSplat | 24.03  | 0.875 | 0.145 | 524k |
> | DepthSplat-O | 24.27  | 0.881 | 0.140 | 524k |
> | Only Local Fusion  | **28.25**  | 0.9178 | 0.1061 | 374k |
> | Window = 4 | *28.15*  | **0.9183** | *0.1060* | **285k** |
> | Window = 8 | 28.03  | *0.9180* | **0.1053** | *314k* |
>
> | 50 Views | PSNR    | SSIM   | LPIPS  | # Gaussians |
> |--|--|--|--|--|
> | DepthSplat | 19.48  | 0.706 | 0.255 | 3277k |
> | DepthSplat-O | 19.96  | 0.727 | 0.246 | 3266k |
> | Only Local Fusion  | 23.145 | 0.799 | 0.186 | 1604k |
> | Window = 4 | **23.434** | **0.810** | **0.181** | **1070k** |
> | Window = 8 | *23.201* | *0.804* | *0.183* | *1493k* |

---

> ### Author Response · Authors · 2025-11-25
> **Thank you for your reviews (Part 2/2)**
>
> **Q4: Unposed-image settings**
>
> **A4:** We agree that our current method is a post-hoc module built on top of a posed feed-forward backbone and therefore inherits its assumptions, including the availability of camera poses. Designing a fully feed-forward pipeline that estimates poses on-the-fly from purely RGB inputs, while simultaneously maintaining high-quality 3DGS reconstruction, is an active and challenging research problem, and is closely related to recent progress in 3D foundation models (e.g., VGGT [A], DepthAnything V3 [B]). We consider this direction complementary to our work rather than the main focus of this paper.
>
> We would like to highlight that our LOGAussian itself is backbone-agnostic and the module itself does not specify how poses are obtained and does not modify the pose-estimation process. Once an unposed or partially posed backbone provides per-frame camera parameters and 3DGS, LOGAussian can in principle be applied on top in the same way. We will futher clarify this limitation in the paper and view combining LOGAussian with unposed 3D backbones as an interesting direction for future exploration.
>
> [A] Wang, Jianyuan, et al. "Vggt: Visual geometry grounded transformer." Proceedings of the Computer Vision and Pattern Recognition Conference. 2025.
> [B] Lin, Haotong, et al. "Depth Anything 3: Recovering the Visual Space from Any Views." arXiv preprint arXiv:2511.10647 (2025).
>
> **Q5: Can LOGAussian handle sequences with thousands of views?**
>
> **A5:** In terms of running GPU memory, our method is not directly tied to the sequence length, thanks to the locality-preserving pruning and updating scheme. In our current experiments, we have tested sequences of roughly **300 frames** (Table G) and did not observe GPU memory becoming the bottleneck. This contrasts with offline feed-forward methods such as Long-LRM [A] and ZPressor [B], which explicitly mention memory and scalability limitations as the number of views increases in their limitation discussions.
>
> The main limitation we anticipate on very long or more challenging videos comes from the **feed-forward backbones** trained under sparse-view regimes, rather than from the fusion module itself. Long image sequences often contain unreliable frames (e.g., textureless regions, large exposure changes, imperfect epipolar geometry, motion blur), under which the depth and feature predictions degrade and most feed-forward 3DGS pipelines suffer.
>
> Since LOGAussian is backbone-agnostic, it can in principle be combined with stronger or more robust backbones to better handle such difficult long-sequence cases. We will clarify in the paper that (1) our current results demonstrate stable memory/runtime behavior up to a few hundred frames, and (2) reconstruction quality on very long videos (e.g., thousands of views) is largely influenced by the chosen backbone.
>
> [A] Ziwen, Chen, et al. "Long-lrm: Long-sequence large reconstruction model for wide-coverage gaussian splats." Proceedings of the IEEE/CVF International Conference on Computer Vision. 2025.
> [B] Wang, Weijie, et al. "ZPressor: Bottleneck-Aware Compression for Scalable Feed-Forward 3DGS." NeurIPS 2025.
>
> **Table G. Performance under roughly 300 frames.**
> | | PSNR | SSIM | LPIPS | Peak GPU memory (GB) | # Gaussians |
> |--|--|--|--|--|--|
> | DepthSplat | OOM | | | | |
> | Ours | 18.0847 | 0.725 | 0.306 | 17 | 3959.4K | |
>
> **Q6: Integration with MVSplat**
>
> **A6:** We thank the reviewer for suggesting to evaluate integration with MVSplat and for the opportunity to further demonstrate that our method is a plug-in module for other feed-forward 3DGS methods. We integrate LOGAussian with MVSplat and evaluate on RE10K under 4-, 8-, and 50-view settings. We also include the original (offline) MVSplat as a reference. Results are summarized in Table F.
>
> **Table F. Integration with MVSplat on RE10K.**
> | 4 views | PSNR | SSIM | LPIPS | # Gaussians |
> |--|--|--|--|--|
> | MVSplat | 23.91 | 0.885 | 0.141 | 262K|
> | + Ours | 27.82 | 0.896 | 0.125 | 176K|
>
> | 8 views | PSNR | SSIM | LPIPS | # Gaussians |
> |--|--|--|--|--|
> | MVSplat | 21.39 | 0.846 | 0.183 | 524K|
> | + Ours | 25.70 | 0.854 | 0.164 | 310K|
>
> | 50 views | PSNR | SSIM | LPIPS | # Gaussians |
> |--|--|--|--|--|
> | MVSplat | OOM | | | |
> | + Ours | 20.96 | 0.707 | 0.261 | 1371K|
>
> LOGAussian consistently improves the rendering quality and reduces the number of Gaussians when plugged into MVSplat, under both 4- and 8-view settings. For 50 views, the original MVSplat runs out of memory (OOM), while the integrated variant remains trainable and produces valid reconstructions. These results support our claim that LOGAussian is a generic, plug-in module that can be combined with existing feed-forward 3DGS pipelines to improve efficiency and robustness.

---

### Official Review · Reviewer_8iHp · 2025-11-01

**Soundness:** 3
**Presentation:** 3
**Contribution:** 3
**Rating:** 6
**Confidence:** 4

**Summary:**

LOGAussian introduces a lightweight module that makes feed-forward 3D Gaussian Splatting (3DGS) suitable for online, real-time scene reconstruction. Traditional 3DGS methods process fixed image sets offline, leading to redundancy and drift when handling streaming inputs. LOGAussian addresses this by incrementally fusing new frames into a consistent global 3D Gaussian model using a local correlation-aware attention mechanism. It combines 2D projections and 3D spatial serialization to detect redundant or inconsistent splats, pruning and refining them efficiently with minimal computational overhead. With about 1% additional parameters, LOGAussian yields a compact, consistent set of splats while maintaining or improving rendering quality as the stream progresses

**Strengths:**

The method achieves a significant reduction in the number of Gaussians used to represent the scene even using the local correspondence matching between Gaussians and the global fusion further decreases the Gaussian count.
Cross dataset generalization is also shown with 12 image sequences and 100 frame evaluations.

**Weaknesses:**

- In section 4.3 a sliding window of M frames is described to perform the global fusion periodically. But the details about M are not obvious.

- Compute used at inference and measures like inference time and GPU memory usage could add further value on top of the number of Gaussians reported.

- The serialization design choices like patch size remain unablated.

- In lines 414-417 perhaps it would be useful to quantify between depthsplat-o and depthsplat-or and the former actually slightly improves over depthsplat as reported in table.2.

- Please follow the submission format, the citation format seems incorrect.

**Questions:**

- Could the authors quantify floaters based on their importance based contribution to the final pixel color and then compare how many of them are removed via the proposed method.

- In equation 5 does the residual update include all Gaussian parameters and can it be quantified in any way perhaps where the most updated features can be identified and their impact on quality isolated.

---

> ### Author Response · Authors · 2025-11-25
> **Thank you for your reviews. (Part 1)**
>
> **Q1 & Q2: Ablation on the sliding window size; inference time and memory usage.**
> **A1 & A2:** Thank you for these suggestions. We have added the detailed efficiency analysis (inference time, FPS, and peak GPU memory) and sliding window size ablations in the general response A1 and below Table B, and will incorporate them into the revised paper.
>
> Our default configuration uses a sliding window size of **M = 4**. We conduct ablations under 8- and 50-view settings, comparing: (i) no global fusion ("Only Local Fusion"), (ii) window size 4, and (iii) window size 8. We also include the offline and online variants of DepthSplat as references. Results are summarized in Table B.
>
> Without global fusion ("Only Local Fusion"), the number of Gaussians remains relatively large, which degrades performance on long image sequences. Adding global fusion reduces the Gaussian count and maintains or improves rendering quality, especially at 50 views. When increasing the window size from 4 to 8, we observe a slight rendering quality drop and less effective pruning. This aligns with our design assumption that Gaussians exhibit locality: the 2D partitioning–based self-attention depends on the number of views, and a larger window makes the fusion more complex and less localized.
>
> **Table B. Ablation on sliding window size M.**
> | 8 Views | PSNR   | SSIM  | LPIPS | # Gaussians |
> |--|--|--|--|--|
> | DepthSplat | 24.03  | 0.875 | 0.145 | 524k |
> | DepthSplat-O | 24.27  | 0.881 | 0.140 | 524k |
> | Only Local Fusion  | **28.25**  | 0.9178 | 0.1061 | 374k |
> | Window = 4 | *28.15*  | **0.9183** | *0.1060* | **285k** |
> | Window = 8 | 28.03  | *0.9180* | **0.1053** | *314k* |
>
> | 50 Views | PSNR    | SSIM   | LPIPS  | # Gaussians |
> |--|--|--|--|--|
> | DepthSplat | 19.48  | 0.706 | 0.255 | 3277k |
> | DepthSplat-O | 19.96  | 0.727 | 0.246 | 3266k |
> | Only Local Fusion  | 23.145 | 0.799 | 0.186 | 1604k |
> | Window = 4 | **23.434** | **0.810** | **0.181** | **1070k** |
> | Window = 8 | *23.201* | *0.804* | *0.183* | *1493k* |
>
> **Q3: Ablation studies on the patch size.**
> **A3:** We ablate the effect of different patch sizes in Table C, where the default configuration uses a patch size of 4. With smaller patch sizes, each patch contains fewer candidate points, leading to fewer pruning opportunities and thus a larger number of retained Gaussians. We also observe that smaller patch sizes yield slightly better rendering quality. This suggests that smaller patches group candidates that are closer in the view space, allowing self-attention to operate on more visually coherent neighborhoods, which can benefit the fused representation.
>
> **Table C. Ablation on patch size.**
> | 4 Views | PSNR | SSIM | LPIPS | # Gaussians |
> |--|--|--|--|--|
> | PatchSize = 2| **28.87** | **0.930** | *0.0961* | 188k |
> | PatchSize = 4| *28.76* | *0.928* | **0.0960** | *183k* |
> | PatchSize = 6| 28.57 | 0.926 | 0.099 | **182k** |
>
> | 8 Views | PSNR | SSIM | LPIPS | # Gaussians |
> |--|--|--|--|--|
> | PatchSize = 2| **28.29** | **0.919** | *0.107* | 292k |
> | PatchSize = 4| *28.15* | *0.918* | **0.106** | *285k* |
> | PatchSize = 6| 27.99 | 0.916 | 0.108 | **282k** |
>
> | 50 Views | PSNR | SSIM | LPIPS | # Gaussians |
> |--|--|--|--|--|
> | PatchSize = 2| **23.489** | 0.809| 0.184| 1099k |
> | PatchSize = 4| *23.434* | **0.810**| **0.181**| *1070k* |
> | PatchSize = 6| 23.426 | **0.810**| *0.182*| **1050k** |
>
> **Q4: Quantifying DepthSplat-O vs DepthSplat-OR; formatting issue.**
> **A4:** Thank you for pointing this out. We will explicitly quantify the differences between DepthSplat-O and DepthSplat-OR in lines 414–417, and fix the corresponding formatting issues in the revised version.

---

> ### Author Response · Authors · 2025-11-25
> **Thank you for your reviews Part (2/2)**
>
> **Q5: Quantify floaters based on their importance-based contribution**
> **A5:** We notice that raw contribution to the final pixel color (e.g., rendering weights) alone is not sufficient to define a "floater" in the feed-forward setting: unlike per-scene optimized 3DGS, feed-forward 3DGS can instantiate Gaussians that contribute strongly to the final pixel colors but exhibit rendering errors. Instead, we use the **gradient norm of the photometric loss w.r.t. the color parameters** as an indicator of how much changing a Gaussian’s color would affect the error.
>
> Concretely, we compute this gradient norm per Gaussian, sort Gaussians by this value, and partition them into five bands (top 20%, 20–40%, …, last 20%). For each band, we measure the **pruning ratio**, i.e., the fraction of Gaussians in that band that are removed by LOGAussian. We run this analysis on 10 scenes with 4 input views and report the averaged pruning ratios in Table H.
>
> **Table H. Pruning ratio across gradient-based importance bands (4 views, averaged over 10 scenes).**
> | Gradient band (by norm w.r.t. color) | Pruning ratio |
> |-|-|
> |Top 20%| 28.13% |
> |20% - 40%| 25.32% |
> |40% - 60% |23.80% |
> |60% - 80% | 25.26% |
> |Last 20%| 31.89% |
>
> We observe that both the **top** and **bottom** 20% bands exhibit higher pruning ratios than the middle bands. Intuitively, the top band corresponds to Gaussians with large gradients (strong influence on the photometric error), many of which behave like floaters or harmful artifacts, while the last band corresponds to Gaussians with very small gradients that are redundant or largely inactive. This suggests that LOGAussian tends to remove both highly erroneous and redundant Gaussians, while being more conservative in the middle bands. We will add this analysis and explanation to the revised paper.
>
>
> **Q6: Analysis on the residual updates**
> **A6:** The residual branch in Equation (5) is defined over **all** Gaussian attributes (means, scales, rotations, opacities, and SH). Because these attributes live in different units and are coupled during optimization, directly comparing raw residual magnitudes is not very informative. Instead, we quantify their impact by **disabling residual updates for specific attributes** and measuring the resulting quality, as shown in Table I (experiments conducted under 8 views).
>
> - "Ours (prune only)" disables **all** residual updates and uses only pruning/fusion on top of DepthSplat-O.
> - Each row with a specific attribute in the "W/O updating" column means that all other attributes are still updated, except that attribute.
> - "Ours (default)" updates **all** attributes jointly.
>
> We observe that all configurations with residual updates significantly outperform DepthSplat-O and the "prune only" variant. Relative to the default, **disabling scales or SH** noticeably degrades PSNR/SSIM and LPIPS, and **disabling opacities** yields slightly higher PSNR but clearly worse SSIM/LPIPS, indicating that opacity corrections are particularly important for perceptual quality. Disabling means or rotations has only a minor effect (differences < 0.03 dB PSNR and very small SSIM/LPIPS changes), suggesting that these attributes are somewhat more robust to being kept fixed in the residual branch.
>
> We hypothesize that this behavior stems from how the model reallocates coverage after pruning: once redundant Gaussians are removed, the **remaining** Gaussians need to adjust their spatial extent (scales) and contributions (opacities and colors) to better cover the scene, which makes residual updates on these attributes especially impactful. During training, we always optimize all attributes jointly and do not train attribute-specific models; this ablation is only used post hoc to analyze their relative influence.
>
> **Table I. Ablation on the residual updates.**
> | Method| W/O Updating | PSNR  | SSIM   | LPIPS  |
> |-|-|-|-|-|
> | DepthSplat-O | – | 24.27 | 0.881 | 0.140 |
> | Ours (prun only) | ALL | 26.23 | 0.904 | 0.125 |
> | Ours | Scales | 27.89 | 0.917 | 0.108 |
> | Ours | Rots | 28.13 | **0.918** | **0.106** |
> | Ours | Opacities | **28.23** | 0.910 | 0.116 |
> | Ours | SH  | 27.76 | 0.916 | 0.111 |
> | Ours | means | *28.15* | **0.918** | 0.107 |
> | Ours (default) | - | *28.15* | **0.918** | **0.106** |

---

### Author Response · Authors · 2025-11-25
**General Response**

We sincerely thank all reviewers for their helpful and constructive feedback. LOGAussian is recognized as tackling a well-motivated problem (MysD) in making feed-forward 3DGS suitable for online / incremental processing (8iHp, MysD, axVZ), with an effective (MysD, 1oHa) and simple / intuitive (MysD, axVZ) design to remove redundant Gaussians (8iHp, 1oHa).
We also thank the reviewers for characterizing our method as a plug-in module to many other feed-forward 3DGS methods (1oHa), with lightweight parameters (8iHp, MysD, 1oHa), which is considered "an important contribution to the community" (1oHa) and "may inspire minor future extensions in online fusion" (axVZ). Experimentally, reviewers commented that our method greatly / consistently improves performance over DepthSplat across datasets (MysD, 1oHa), and shows cross-dataset generalization (8iHp).

In the following, we first address common questions raised across reviews, and then respond to individual comments in detail.

## General Response

**Q1: Inference time and peak GPU memory**
**A1:** In Table A, we report end-to-end inference time, peak GPU memory usage, and rendering FPS for three input-view configurations (4, 8, and 50 views). We compare our method against MVSplat (offline), DepthSplat (offline) and its online variant (DepthSplat-O). Across all settings, our approach consistently achieves higher FPS than baselines. Compared to the offline DepthSplat, our method uses substantially less memory and scales more favorably as the number of input views increases. Compared to DepthSplat-O, our method incurs only modest runtime/memory overhead for few views, and becomes both faster and more memory-efficient at 50 views due to effective pruning. MVSplat runs out of memory (OOM) at 50 views. We also would like to highlight that our approach achieves better rendering quality than all baseline methods by a large margin as reflected by Table 1 and 2(a) in the main paper.

**Table A. Inference time, peak GPU memory, and FPS under different numbers of input views.**
| 4 Views | Inference Time (ms) | GPU Memory (GB) | Rendering FPS |
|--|--|--|--|
| MVSplat | 102.70 | *2.48* | 460 |
| DepthSplat | *71.11* | 4.77 | *489* |
| DepthSplat-O | **39.14** | **1.70** | 462 |
| Ours | 75.35 | 2.82 | **531** |

| 8 Views | Inference Time (ms) | GPU Memory (GB) | Rendering FPS |
|--|--|--|--|
| MVSplat | 218.13 | 4.9 | 368 |
| DepthSplat | 117.97 | 9.25 | 394 |
| DepthSplat-O | **45.36** | **2.06** | *398* |
| Ours | *86.74* | *3.32* | **455** |

| 50 Views | Inference Time (ms) | GPU Memory (GB) | Rendering FPS |
|--|--|--|--|
| MVSplat | OOM | | |
| DepthSplat | 576.56 | 56.29 | 167 |
| DepthSplat-O | **51.35** | *9.11* | *171* |
| Ours | *63.25* | **7.25** | **298** |

---

### Meta-Review · Area_Chair_MP9N · 2025-12-29

**Summary:**

The reviewers recognized the motivation of LOGAussian in adapting feed-forward 3D Gaussian Splatting for online and streaming reconstruction. The paper’s lightweight approach to reducing redundancy in streaming scenarios was acknowledged by reviewers, but the two negative review recommendations are primarily due to the lack of implementation details and comprehensive evaluations. While the rebuttal provided additional metrics and ablation studies on hyperparameters, it did not fully resolve fundamental concerns regarding the method's long-term stability in thousand-frame sequences or its reliance on posed input. Consequently, while the proposed pipeline is intuitive, it might be currently viewed as a marginal extension, and more rigorous evaluations and a deeper exploration of long-term drift are suggested.

**Reviewer Concerns:**

The authors have addressed several concerns in their rebuttal:
- Implementation details: the authors clarified the default settings for the pruning threshold $\tau$ and the sliding window size $M$.
- Computational budgets: in Table A, the authors provided a thorough evaluation of inference speed, peak GPU memory, and rendering FPS across multiple view configurations.
- Ablation of design choices: new ablations on patch size and the residual update branch (SH, scale, etc.) provided an understanding of the module’s sensitivity to specific Gaussian attributes.
- Pruning mechanism: the gradient-based importance analysis helped quantify how the module targets redundant or harmful GS versus high-contribution splats.

However, two concerns are still outstanding:
- Scalability to >1k input frames: while results for ~300 frames were provided, it remains unclear whether the current local-gathering mechanism can sufficiently mitigate error accumulation and memory bound in very long sequences (thousands of frames).
- Reliance on input poses: the framework’s dependency on precise, pre-computed poses limits its applicability in pose-free or autonomous scenarios. As noted by the reviewers, the requirement for an external real-time SLAM system introduces more complexity and overhead that potentially negates the benefits of a "lightweight" streaming design.

**Reviewer Scores:**

The manuscript first received a mixed initial review score of (6, 4, 6, 2). After the rebuttal/discussion and before the openreview incident, none of the reviewers had replied or changed their scores.

- Reviewer 8iHp initially gave a positive score of 6 and raised several technical questions that should have been provided in the paper. The authors have addressed them all, but I am not confident that the reviewer would raise the score.
- Reviewer MysD initially gave a score of 4. The authors answer most of the raised questions, but the scalability to >1k input frames and reliance on input poses are still outstanding, so I would assume reviewer MysD would remain the negative rating.
- Reviewer 1oHa initially gave a positive score of 6 and mentioned that would raise the score if the authors provide additional experiments on performance given full sequence videos/images. The authors addressed this part in the rebuttal, and I would assume a raised score of 8.
- Reviewer axVZ gave a negative score of 2 and I don't think the rating direction would change after the rebuttal (especially given the concerns on posed input)

Based on the above analysis, I would assume a final score around (6, 4, 8, 2). Considering also the outstanding concerns on scalability and posed input, I cannot support the manuscript given its current outlook.

---

### Decision · Program_Chairs · 2026-01-26

Reject